# Risk Factors for Positive Appraisal of Mistreatment during Childbirth among Ethiopian Midwifery Students

**DOI:** 10.3390/ijerph17082682

**Published:** 2020-04-14

**Authors:** Rena Bakker, Ephrem D. Sheferaw, Tegbar Yigzaw, Jelle Stekelenburg, Marlou L. A. de Kroon

**Affiliations:** 1Department of Health Sciences, Global Health, University Medical Center Groningen, University of Groningen, 9713 AV Groningen, The Netherlands; e.d.sheferaw@rug.nl (E.D.S.); j.stekelenburg@umcg.nl (J.S.); 2Jhpiego Ethiopia, Addis Ababa, Ethiopia; tegbar.yigzaw@jhpiego.org; 3Department of Obstetrics and Gynecology, Leeuwarden Medical Centre, 8934 AD Leeuwarden, The Netherlands; 4Department of Health Sciences, University Medical Center Groningen, University of Groningen, 9713 AV Groningen, The Netherlands; m.l.a.de.kroon@umcg.nl

**Keywords:** disrespect and abuse, respectful maternity care, midwifery students, Ethiopia

## Abstract

The maternal mortality ratio and neonatal mortality rate remain high in Ethiopia, where few births are attended by qualified healthcare staff. This is partly due to care providers’ mistreatment of women during childbirth, which creates a culture of anxiety that decreases the use of healthcare services. This study employed a cross-sectional design to identify risk factors for positive appraisal of mistreatment during childbirth. We asked 391 Ethiopian final year midwifery students to complete a paper-and-pen questionnaire assessing background characteristics, prior observation of mistreatment during education, self-esteem, stress, and mistreatment appraisal. A multivariable linear regression analysis indicated age (*p* = 0.005), stress (*p* = 0.019), and previous observation of mistreatment during education (*p* < 0.001) to be significantly associated with mistreatment appraisal. Younger students, stressed students, and students that had observed more mistreatment during their education reported more positive mistreatment appraisal. No significant association was observed for origin (*p* = 0.373) and self-esteem (*p* = 0.445). Findings can be utilized to develop educational interventions that counteract mistreatment during childbirth in the Ethiopian context.

## 1. Introduction

Mistreatment during childbirth is a human rights violation in which disrespectful care provision is inflicted upon childbearing women, while their wishes and needs are neglected [1,2]. It may encompass malpractices such as physical abuse, verbal abuse, non-consented care, non-confidential care, discrimination based on patient attributes, abandonment of care, and detention in facilities [3]. Mistreatment during childbirth is an issue across the globe, yet its occurrence is particularly prevalent in low-income settings [4]. Factors such as frustration among healthcare personnel and unequal patient–provider relations can help to explain why women are being mistreated during childbirth; however, intrinsically good motives among healthcare staff might play a role as well [5]. This can be well understood through the words of an Ethiopian midwife.

They do that [abuse] for the sake of the mothers. When the labor is in the second stage, and the mother doesn’t care for the baby, the midwives may slap the thigh of the mother only with the aim to save the baby[6] (p. 9).

Irrespective of possibly well-meant intentions, mistreatment during childbirth ultimately creates a culture of fear that reduces pregnant women’s healthcare utilization [7]. As maternal healthcare utilization is a major predictor of mother and child well-being, mistreatment during childbirth eventually contributes toward maternal and neonatal morbidity and mortality [8]. It hence endangers patient safety, while increasing the risk of preventable adverse events [9].

In order to improve maternal and child well-being and reach the United Nations Sustainable Development Goal 3, which aims to reduce the global maternal mortality ratio to 70 per 100,000 live births and the neonatal mortality rate to 12 deaths per 1000 live births, a more patient-centered care approach is needed [2,10,11,12]. This is also emphasized by the Respectful Maternity Care (RMC) framework, which promotes women’s autonomy and dignity in childbirth [13]. Next to mobilizing communities to demand RMC, along with integrating RMC into the curriculum for maternity healthcare providers and supporting these healthcare providers to act accordingly, RMC should ultimately be integrated into national legislation and healthcare policies [14].

For the development and implementation of effective RMC interventions, it is essential to gain a better understanding of the etiology of mistreatment during childbirth. Research has indicated mistreatment during childbirth to stem from societal level risk factors (e.g., cultural beliefs), organizational level risk factors (e.g., conditions at work), and individual level risk factors among healthcare staff (e.g., beliefs and personal attributes), which may then give rise to poor health service characteristics and inadequate interpersonal interactions between healthcare staff and patients [15]. While factual knowledge is an important pillar of good communication in the healthcare branch, proper acknowledgment of interpersonal and relational information is a critical skill among midwives and obstetricians, as childbirth is often associated with feelings of anxiety and uncertainty [9].

Previous literature on mistreatment during childbirth and individual level risk factors among care providers is limited, yet some characteristics have been outlined, such as young age, observation of mistreatment during education, low self-esteem, and high stress levels [6,15,16,17,18]. It is important to acknowledge that these findings stem from different countries and sample populations, where not all factors might play a role in each setting. Different norms and traditions in diverse settings, variations regarding education and midwifery training, as well as dissimilarities related to health facilities’ condition and equipment are likely to affect care providers’ competencies, attitudes, and interactions with patients [19,20,21]. In order to develop and promote effective RMC interventions in a particular setting, these factors underline the value of context-specific research on risk factors for mistreatment during childbirth.

Ethiopia constitutes the basis for our investigations, as mistreatment during childbirth is a frequently encountered problem in this country, where the maternal mortality ratio and the neonatal mortality rates are high [21]. An important step in advocating RMC in Ethiopia encompasses adjusting the curriculum of health science programs and promoting RMC among midwifery students [6,21,22]. Therefore, it is essential to understand which students are at risk for mistreating women during childbirth.

Previous research on individual level risk factors for mistreatment during childbirth in Ethiopia is scarce; research has been mostly conducted among care providers, while studies among midwifery students are extremely scarce, and findings have been inconsistent [6,23]. One study has for example pointed toward more RMC among male Ethiopian healthcare providers. However, in our earlier published study, gender did not predict positive appraisal of mistreatment during childbirth among Ethiopian midwifery students [24]. The lack and inconclusiveness of previous research led to the aim of the current study, which was to analyze and identify risk factors for mistreatment during childbirth among Ethiopian midwifery students.

## 2. Materials and Methods

This study employed a cross-sectional design. Local data collectors asked Ethiopian final-year midwifery students from six educational institutions (i.e., Gondar University, Bahir Dar University, Bahir Dar Health Science College (HSC), Hawassa HSC, Arsi University, and Menelik HSC) to participate in this study during class. Data were collected from November to December 2017 and the response rate was 100%. We excluded one student from the analysis due to their completing an English version and thus not an Ethiopian version of the questionnaire. The final study sample included 391 students. The study protocol was reviewed and approved by the Johns Hopkins Bloomberg School of Public Health Institutional Review Board (IRB00008218). All students gave informed consent before participating.

Students were asked to complete a paper-and-pen questionnaire that captured various background characteristics, observation of mistreatment during education, self-esteem, stress, and appraisal of mistreatment. Self-esteem and stress were assessed using previously translated Amharic versions of the Rosenberg Self-Esteem Scale and Perceived Stress Scale that showed acceptable degrees of internal consistency (α = 0.73 and α = 0.76, respectively) [25,26]. All other questions underwent a forward and backward translation process, carried out by Ethiopian epidemiology master students from the University of Groningen, the Netherlands.

The *outcome variable* was appraisal of mistreatment during childbirth (α = 0.75), and it was assessed with the 10-item Mistreatment Appraisal (MISAP) Scale [24]. The items of the MISAP scale are based on the typology of mistreatment by Bohren et al. [4] and encompass the following mistreatment themes: physical abuse, verbal abuse, stigma and discrimination, failure to meet professional standards of care, poor rapport between women and providers, and health system conditions and constraints. Individuals were asked to rate actions that depict mistreatment during childbirth on a scale ranging from 1 (*oppose strongly*) to 10 (*support strongly*). Solely endpoints, and hence no in-between numbers of the scale, received a label. This gave rise to theoretical outcome scores of 10 to 100, with higher scores depicting more positive appraisal of mistreatment during childbirth. We allowed 20% of items to be missing, which was the case in 6% (N = 25). When data were missing, the weighted mean sum scores were calculated.

The *independent variables* were place of origin (urban, rural), age, observation of mistreatment during education, self-esteem, and stress. To assess observation of mistreatment during education (α = 0.70), we used ten items of a questionnaire by Moyer et al. [17]. Students had to indicate on a scale, including: 1 (*never*), 2 (*rarely*), 3 (*sometimes*), and 4 (*most of the time*), whether they had observed disrespectful maternity care during their education thus far. Theoretical outcome scores ranged from 10 to 40. Higher scores depicted more observation of mistreatment during education. Self-esteem (α = 0.61) was assessed with the 10-item Rosenberg Self-Esteem Scale [27]. This scale included the labels: 1 (*strongly agree*), 2 (*agree*), 3 (*disagree*), and 4 (*strongly disagree*), and gave rise to theoretical outcome scores of 10 to 40. Higher scores indicated more self-esteem. One item *(I wish I could have more respect for myself)* was removed due to its ambiguous wording, which improved the scale’s internal consistency (α = 0.72), yielding theoretical outcome scores of 9 to 36. Stress (α = 0.71) was measured with the 10-item Perceived Stress Scale [28]. This scale included the labels: 0 *(never)*, 1 *(almost never),* 2 *(sometimes)*, 3 *(fairly often),* and 4 *(very often)* and yielded theoretical outcome scores of 0 to 40, with higher scores depicting more stress. We allowed 20% of observation of mistreatment, self-esteem, and stress items to be missing, which was the case in 19% (N = 74), 7% (N = 27), and 6% (N = 25), of the items respectively. When data were missing, the weighted mean sum scores were calculated.

*Covariates* were gender (male, female), institution (Gondar University, Bahir Dar University, Bahir Dar HSC, Hawassa HSC, Arsi University, or Menelik HSC), religion (Orthodox, Protestant, Islam, or other), and ethnicity (Amhara, Oromo, or other).

In total, 1% of all values were missing (i.e., 44 values). To reduce the impact of missing data, we used multiple imputation to generate five datasets for our main analyses. Data imputation was conducted for gender (1 value), place of origin (1 value), age (21 values), observation of mistreatment during education sum scores (8 values), stress sum scores (7 values), and appraisal of mistreatment sum scores (6 values). Imputed values were sampled from a predictive distribution based on the associations between all covariates, independent, and outcome variables [29].

Background characteristics, independent variables, and the outcome variable were described by means, standard deviations (SD), ranges and/or counts, and percentages. After examining the assumptions of linearity, we conducted a correlational analysis of our cross-sectional data set using multiple linear regression analyses to assess the association between appraisal of mistreatment and place of origin, age, observation of mistreatment during education, self-esteem, and stress, while controlling for gender, institution, religion, and ethnicity [30]. Variables that yielded *p* < 0.25 when applying bivariable linear regression analysis were added to the multivariable regression analysis simultaneously [31]. Analyses were conducted with SPSS statistical software version 25 for Windows (SPSS Inc. Chicago, IL, USA). *p*-values < 0.05 were considered significant.

## 3. Results

Table 1 shows the background characteristics of the population. Most individuals were female students from Gondar University, with an orthodox, urban background from the Amhara region. The mean age was 23.58 years (SD 2.56, range 20.00–40.00). Observation of mistreatment scores were moderate (mean 19.71, SD 5.14, range 11.00–37.00). Students had a rather high self-esteem (mean 29.40, SD 4.27, range 13.00–36.00) and moderate stress levels (mean 15.24, SD 5.71, range 1.00–34.00), while appraisal of mistreatment scores were rather low (mean 33.92, SD 15.06, range 10.00–90.00).

Table 2 shows that place of origin, age, observation of mistreatment during education, self-esteem, and stress were all significantly related to appraisal of mistreatment in a bivariable regression analysis. Table 2 also shows that age, observation of mistreatment during education and stress remain significantly related to appraisal of mistreatment in a multivariable regression analysis; while adjusting for gender, institution, religion and ethnicity, with more positive appraisal of mistreatment during childbirth among younger students, students that observed more mistreatment during education and stressed students.

## 4. Discussion

Our study examined individual level risk factors for mistreatment during childbirth among Ethiopian midwifery students, and it showed that there was a significant association between positive appraisal of mistreatment and younger age, observation of mistreatment during education, and stress. However, our study did not confirm that place of origin and self-esteem were associated with appraisal of mistreatment. These results are partly in line with previous research that has highlighted the importance of interpersonal interactions among healthcare staff and patients herein, and various individual level risk factors among care providers, such as young age, observation of mistreatment during education, low self-esteem, and high stress levels [6,9,15,16,17,18]. The significant inverse relationship between positive appraisal of mistreatment and age, is in line with previous studies that have pointed toward more patient-centeredness and better communication skills among older students [32,33,34]. Possibly, the less positive appraisal of mistreatment among older midwifery students can be attributed to factors such as life experience (e.g., having delivered themselves or witnessed their wives give birth), which may encompass giving older students a more critical attitude and assertive manner that might make them less prone to copy negative behavior from their instructors [32,33,35].

Our finding that observation of mistreatment during education was significantly related to positive appraisal of mistreatment, confirms findings of Rominski et al. [5], who proposed that observation of mistreatment during education might cause midwifery students to rationalize disciplinary measures against patients through victim-blaming. While students might understand the importance of respectful and patient-centered care, observation of mistreatment during education appears to put them at risk of internalizing such behaviors. In turn, the normalization of mistreatment during childbirth and the structural embeddedness of violence in the healthcare setting underlines the complexity of promoting behavioral change, as it requires fundamental individual level, organizational level, and societal level transformations [15,36,37].

The statistically significant relationship between stress and positive appraisal of mistreatment, confirms previous research findings that have identified stress as a root cause of aggression due to diminished cognitive processing and self-regulation [18,38]. Healthcare personnel across the globe are exposed to high job demands; however, poor regulatory frameworks, as well as unsafe and inadequate working conditions, are particularly prevalent in low-income settings, which are likely to induce more stress and disrespectful patient–provider interactions [39].

Despite more conservative norms and more appraisal of gender violence in rural Ethiopian areas, we did not observe an independent relationship between rural origin and a more positive appraisal of mistreatment among midwifery students [19,40]. The fact that midwifery students commonly no longer live in rural communities while undergoing professional training, might account for this insignificant finding. This result is also in line with earlier findings showing that patients’ origin, rather than care providers’ origin, affects their respective risk of being mistreated during childbirth [4]. That is, women from rural areas with a lower socioeconomic status have been reported to receive worse treatment. Finally, rural origin might be highly associated with observation of mistreatment, which was a significant predictor in the multivariable analysis.

An explanation for the finding that self-esteem was not related to appraisal of mistreatment can be derived from the notion that self-esteem is a highly situation-dependent state, rather than a stable trait that persists across different situations [41]. A real-life setting might frequently induce low self-esteem upon students due to complex power dynamics (e.g., a physician and a midwifery student working together). However, as participation in this study cannot capture real-life power dynamics, it is likely that students’ self-esteem was unaffected by such mechanisms. Another explanation for the insignificant association between self-esteem and appraisal of mistreatment, stems from the view that self-esteem is a highly individualistic concept that might be less applicable in the collectivistic Ethiopian setting [42,43]. Moreover, the variables, stress and self-esteem, may be highly correlated, which may account for the finding that self-esteem was not independently associated with the outcome variable.

Our study had a number of strengths and limitations. An important strength of this study was its design. While previous research mostly used qualitative methods, we utilized quantitative measures and sampled data from different locations, which increases the study’s generalizability. Another strength was the data quality, which was ensured by employing local processional data collectors, who approached students during class. Limitations of this study encompass: sampling students from merely four regions, limiting the study’s generalizability; the possibility that social desirability bias may have affected study outcomes, despite guaranteeing anonymity; and its cross-sectional set-up, which implies that no conclusions on causality can be drawn. We recommend that controlled trials or longitudinal studies be performed in the future to gain more insight into causal pathways.

Mistreatment during childbirth is a complex, multi-component problem that requires a thorough understanding of the reasons for its emergence and its perception by different stakeholders. Accordingly, interventions need to focus on various aspects, such as improving the quality of patient–provider interactions and creating accountability for mistreatment during childbirth at an organizational and national level, which implies that there is no “one fits all” solution [44]. Ultimately, mistreatment during childbirth is often linked to limited financial resources, yet associated structural shortcomings are difficult to overcome [3,39]. The present study can contribute toward this issue by providing insights into interventions that particularly affect midwifery education.

With regard to the finding that younger midwifery students were more likely to appraise mistreatment during childbirth positively, it might be beneficial to reconsider admission procedures for health-related studies in Ethiopia. The implementation of a minimum age, and the promotion of secondary entry degrees for studies such as nursing, midwifery, and medicine may contribute toward better patient–provider interactions. In fact, Ethiopia launched its New Medical Education Initiative (NMEI) in 2012, which is a secondary degree medical training program that seeks to diminish the shortage of medical doctors [45]. As NMEI students are typically older than regular students, it would be interesting for future research to assess whether these students are more prone to provide RMC.

The association between negative role modeling behavior of instructors and positive appraisal of mistreatment among students, underlines the importance of raising awareness about disrespectful care provision at higher education institutions. The current Ethiopian technical and vocational education and training model curriculum for midwifery students encompasses a 75 h module (i.e., 2% of the total degree program duration) that promotes RMC provisions [46]. However, once midwifery students do their clinical internships, they will most likely experience a mismatch between the behavior of their instructors and the principles of RMC that they had previously been taught. Negative consequences of this discrepancy could be leveraged by making students aware of the fact that the ideals of RMC are often not attained in practice, but that mistreatment during childbirth is nonetheless unacceptable under any circumstances, next to ensuring good supervision. One practical approach to achieve this goal could encompass the establishment of supervised feedback days, during which students can present and discuss real life situations of mistreatment during childbirth. Additionally, it appears crucial that the RMC framework is integrated more substantially into the Ethiopian midwifery education curriculum.

The observed association between stress and positive appraisal of mistreatment, highlights the importance of strengthening student support facilities, such as counseling services that facilitate effective stress management and coping strategies. As stress exposure is an ongoing issue in the healthcare branch, facilitating counseling for both midwifery students and working midwives is likely to be useful [18,39].

## 5. Conclusions

Promoting RMC among midwifery students comprises an important step in overcoming mistreatment during childbirth [22]. This study points toward a significant association between positive appraisal of mistreatment and younger age, observation of mistreatment during education, and stress among Ethiopian midwifery students. Findings of this study can contribute toward enhancing educational interventions that target these risk factors, and ultimately increase professionalism among Ethiopian midwifery students, who constitute an important proportion of Ethiopia’s future healthcare workforce.

## Figures and Tables

**Table 1 ijerph-17-02682-t001:** Background characteristics of the study population (N = 391).

Variable	N (%)
Gender	
Male	151 (39)
Female	239 (61)
Institution	
Gondar University	130 (33)
Bahir Dar University	38 (10)
Bahir Dar HSC	42 (11)
Hawassa HSC	59 (15)
Arsi University	40 (10)
Menelik HSC	82 (21)
Religion	
Orthodox	304 (78)
Protestant	39 (10)
Islam	42 (11)
Other	6 (1)
Place of origin	
Urban	209 (54)
Rural	181 (46)
Ethnicity	
Amhara	255 (65)
Oromo	66 (17)
Other	70 (18)

**Table 2 ijerph-17-02682-t002:** The relationships of possible risk factors and appraisal of mistreatment in bivariable regression and multivariable regression analyses.

Variable	Bivariable Linear Regression			Multivariable Linear Regression ^1^		
	*B*	95% CI	*p*	*B*	95% CI	*p*
Place of origin (urban is reference category)	−4.202	[−7.246, −1.159]	0.007	−1.488	[−4.766, 1.790]	0.373
Age	−1.078	[−1.675, −0.481]	<0.001	−0.851	[−1.445, −0.256]	0.005
Observation of mistreatment	0.775	[0.492, 1.058]	<0.001	0.549	[0.264, 0.833]	<0.001
Self-esteem	−0.731	[−1.082, −0.380]	<0.001	−0.155	[−0.554, 0.244]	0.445
Stress	0.584	[0.320, 0.848]	<0.001	0.356	[0.060, 0.652]	0.019

^1^ Adjusted for gender, institution, religion, and ethnicity.

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
