# Peer review of "Risk Factors for Positive Appraisal of Mistreatment during Childbirth among Ethiopian Midwifery Students"

_ijerph, 2020, doi:10.3390/ijerph17082682_

Round 1

Reviewer 1 Report

Very interesting topic and your paper was nicely written without many errors. There were a few minor issues to correct:

  1.  in text editing before publication
  2. create an appendix section for scale ranges or expand to all of the ranges instead of writing.

Line 25-need a comma before "and"

Line 28-need a comma before "and"

Line 70-rates instead of rate

Lines, 99, 109, 111, 113-List all of the scale ranges or create an appendix section to list them all.

Line 164-change to e.g., instead of e.g.

Line 171-victim blaming should be victim-blaming and multicomponent should be multi-component

Line 221-delete "a"

Put your www. Grammarly.com to ensure punctuation is being correct.

Reviewer 2 Report

Bakker et al in their report studied the behavioral issue of midwifery students towards the clients. The study involves sufficient number of subjects to derive the meaningful conclusion. The data suggest age, stress and previous mistreatment observation were the key factors responsible for the behavior. The study raised important question in context to Ethiopia.

The study is well within the scope of the journal and may be accepted for the publication after minor revision

  1. Table 1: Religion parameters should be removed from the table. It is not wise to mix religion with science.
  2. How does the data derived from the study can be used by policy makers, should be discussed?

Reviewer 3 Report

The manuscript addresses an important question on what which risk factors are relevant for mistreatment during childbirth among Ethiopian midwifery students when controlling for each other and further covariates. However, there many limitations which need to be addressed in a revision:

1) Mistreatment during childbirth needs to be explained with more detail as the reader must be able to understand what behavior this is actually. Please relate to patient safety and preventable adverse events (pAEs).

2) A theoretical backdrop is missing and needs to be added and linked explicitly already in the introduction and also to the discussion section. 

3) The 10-item Mistreatment Appraisal (MISAP) Scale needs to be reported in detail, and a “compliance” rate should be calculated in terms of identifying a group that is above the threshold of showing mistreatment behavior.

4) Was ethics approval obtained? Did all study participants consent to participate in the study? If not, this needs to be obtained at least post-hoc to allow publication.

5) This appears to be a correlational analysis of cross-sectional data set. While this is fine for a needs-assesses, the explicit mention of this should be added.

6) Please answer all your research questions and aims, and relate back to the theoretical basis of this study.

Please feel free to look into and cite Lippke, S., Wienert, J., Keller, F.M., Derksen, C., Welp, A., Kötting, L., Hofreuter-Gätgens, K., Müller, H., Louwen, F., Weigand, M. Ernst, K., Kraft, K., Reister, F., Polasik, A., Huener nee Seemann, B., Jennewein, l., Scholz, C. & Hannawa, A. (2019). Communication and patient safety in gynecology and obstetrics – study protocol of an intervention study. BMC Health Serv Res, 19(1), 908, doi:10.1186/s12913-019-4579-y

Round 2

Reviewer 3 Report

This cross-sectional study is important and provides interesting results. While I see the authors significantly revised their paper, there is still much to be done: 1) A clear theoretical Basis Needs to be added, see for instance https://bmchealthservres.biomedcentral.com/articles/10.1186/s12913-019-4579-y 

2) Research Questions Need to be explicitly mentioned and tested in the results section.

3) The results should be extended by running some more analyses enlightening the theoretical assumptions and to better understand the barriers in the field.

4) The discussion needs to review more critically the shortcomings including the cross-sectional study design. Implications for future studies should also include the generalizability of the findings to other countries and cultures.